# The "Green" FMOs: Diversity, Functionality and Application of Plant Flavoproteins

**Sara Thodberg \* and Elizabeth H. Jakobsen Neilson \***

Plant Biochemistry Laboratory, Department of Plant and Environmental Sciences, University of Copenhagen, Thorvaldsensvej 40, 1871 Frederiksberg C, Denmark

\* Correspondence: sart@plen.ku.dk (S.T.); en@plen.ku.dk (E.H.J.N.)

**Abstract:** Flavin-dependent monooxygenases (FMOs) are ancient enzymes present in all kingdoms of life. FMOs typically catalyze the incorporation of an oxygen atom from molecular oxygen into small molecules. To date, the majority of functional characterization studies have been performed on mammalian, fungal and bacterial FMOs, showing that they play fundamental roles in drug and xenobiotic metabolism. By contrast, our understanding of FMOs across the plant kingdom is very limited, despite plants possessing far greater FMO diversity compared to both bacteria and other multicellular organisms. Here, we review the progress of plant FMO research, with a focus on FMO diversity and functionality. Significantly, of the FMOs characterized to date, they all perform oxygenation reactions that are crucial steps within hormone metabolism, pathogen resistance, signaling and chemical defense. This demonstrates the fundamental role FMOs have within plant metabolism, and presents significant opportunities for future research pursuits and downstream applications.

**Keywords:** Evolution; Flavin-dependent monooxygenase; flavoprotein; FMO; general metabolism monooxygenase; plant; S- and N- hydroxylation; specialized metabolism; systemic acquired resistance; YUCCA

## 1. Introduction

Flavin-dependent monooxygenases (FMOs; EC 1.14.13.8) constitute a large group of ancient enzymes that are present in all kingdoms of life. By the utilization of NAD(P)H, FMOs catalyze the incorporation of an oxygen atom from molecular oxygen into small nucleophilic or electrophilic molecules. The oxidation of these molecules, sometimes with high chemo-, regio- and enantio-selectivity, alters their physical and chemical properties, such as polarity, solubility, reactivity, and susceptibility for further enzymatic modifications [1]. Accordingly, FMOs are important oxidoreductases with large potential for development as biocatalysts within the biotechnological and pharmaceutical industries [1–6]. Traditionally, other oxygenase enzymes, such as cytochromes P450 (CYPs), have received comparatively more attention for their endogenous roles in vivo [7–9], and in regard to the identification and development of new enzyme reactions. This is surprising considering FMOs possess a number of advantageous properties compared to CYPs, especially for downstream biotechnological applications. From a catalytic perspective, FMOs are self-sufficient and do not require additional redox partners or the presence of a substrate for activation [10]. Furthermore, FMOs can have higher catalytic rates compared to CYPs [7], are rarely inhibited by different substrates [10,11], and are soluble enzymes capable of being heterologously expressed in prokaryotic hosts.

Over the last decade, the characterization of FMO function has largely been driven by studies within mammals (human), fungi and bacteria due to highly relevant activities relating to human health and the identification of new biocatalysts for biotechnological applications [12,13]. For example, within

human FMO research, it has been shown that all five active isoforms possess wide substrate specificity, playing a critical role in the detoxification of different xenobiotics [14]. In recent years it has become more apparent that FMOs, such as FMO3 and FMO5, perform many of the detoxification reactions in the liver [15]. Furthermore, FMOs are linked to human health conditions such trimethylaminuria (fish odor syndrome) and atherosclerosis, caused by an FMO3 imbalance. Similarly, FMO1 and FMO5 have also been linked to the prevalence of diabetes, and, more recently, FMOs were shown to be involved in aging and cholesterol homeostasis [7,16–18]. Studies into fungal and bacterial FMOs have identified biotechnologically applicable reactions and enzymatic properties [4], with some bacterial FMOs currently used in the production of dyes and pharmaceuticals [19,20].

In contrast, the last decade of research into plant FMOs has progressed more slowly compared to other kingdoms, despite plants possessing far greater FMO diversity (Figure 1B). In 2007, a detailed review of plant FMOs by Schlaich et al. [21] concluded that for the field to progress "*it is necessary to first identify the endogenous substrates*". This insight has proven to be correct, and, at present, only a small handful of plant FMOs have been characterized and the endogenous substrates for plant FMOs still largely remain unknown.

This review aims to summarize the diversity and role of characterized plant FMOs. Specifically, we will highlight the field's progress since the detailed review by Schlaich in 2007; a review made when the proposed role of YUCCAs and glucosinolate-related FMOs was just established [21]. With the large developments in genome and transcriptome sequencing, this review will also provide an evolutionary context of FMO diversification. Lastly, we seek to highlight the opportunities of FMO research for the forthcoming years to maximize the potential of FMOs for future downstream applications.

## 2. Monooxygenase Enzymes Play Critical Roles in Plant Metabolism

Plants produce an immense number of metabolites that play essential roles in general and specialized metabolism [22]. Monooxygenase enzymes, and the incorporation of oxygen into a molecule is a critical step in the production of both general and specialized metabolites. This incorporation of oxygen alters the metabolite physical and chemical properties such as polarity, solubility, reactivity, and susceptibility for further enzymatic modification such as glycosylation and acetylation.

Plant general metabolites are widely distributed throughout the plant kingdom and are essential to plant growth and reproduction, e.g., hormones, carbohydrates, and amino acids [23]. General metabolites, and their corresponding biosynthetic pathways, hold a certain degree of rigidity, and remain highly conserved, despite millions of years of evolution. In contrast, specialized metabolites play an essential role in how plants evolve and adapt to their environment, such as responding to abiotic stress, the attraction of pollinators, and protection against herbivores and pathogens. These metabolites are often lineage-specific, with diversified and novel biosynthetic pathways.

Interestingly, plants possess a higher number of monooxygenases compared to species in other kingdoms, and are especially diversified regarding major classes of CYPs [24] and the iron/2-oxoglutarate-dependent oxygenases (Fe/2OGs) [6]; with several hundred CYP and Fe/2OG enzymes encoded by genes within a single plant genome. Whilst not as numerous as other monooxygenase classes, FMOs are also highly diversified compared to the number of FMOs present within other kingdoms. For example, humans only possess five FMOs, whilst Arabidopsis (*Arabidopsis thaliana*) and barley (*Hordeum vulgare*) possess 29 and 41 FMOs in their genomes, respectively (Figure 1B). This diversification of plant monooxygenases in general, and FMOs specifically, has largely been attributed to the ability of plants to produce a massive diversity of specialized metabolites, and their plasticity to respond and interact to their abiotic and biotic environment [23].

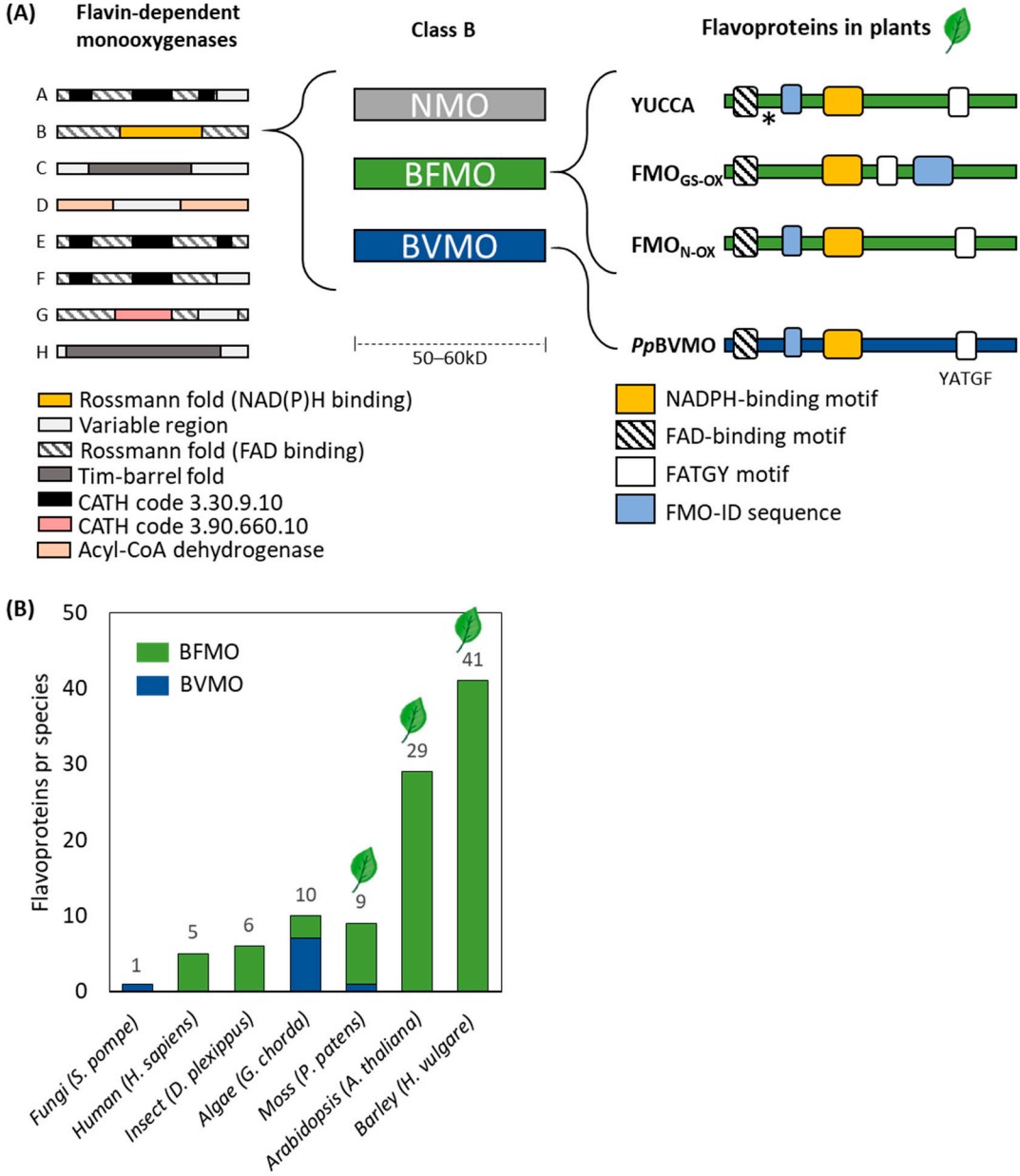

**Figure 1.** An overview of flavin-dependent monooxygenases and flavoproteins in plants. (**A**) Flavin-dependent monooxygenases (FMOs) are separated into eight classes, A–H. Schematic representation of the classes is shown, with corresponding CATH domains folds (modified from [25]). All plants possess Class B FMOs, which consist of two Rossmann folds (CATH code 3.50.50.60), a NADPH binding site (yellow) and a FAD binding site (grey, and split into two domains). Class B flavoproteins consist of three distinct sub-classes: N-hydroxylating monooxygenases (NHMO), Class B Flavoproteins (BFMOs) and Baeyer-Villiger monooxygenases (BVMOs). An overview of the flavoproteins identified in plants illustrates the distribution and alignment of the FAD and NADPH binding sites, the FATGY motif and the FMO-identifier motif in three types of Arabidopsis BFMO proteins, and the *P. patens* BVMO. The asterix denotes a conserved cysteine in Arabidopsis YUCCAs. (**B**) Examples of Class B FMO diversity across kingdoms, with the number of BFMOs selected from single and multicellular organisms as acquired from online resources (see Supplementary Materials DataS1 file for further details).

### 3. Plants possess Class B Flavoproteins

Overall, FMOs are divided into eight different classes; A–H, as defined by their phylogenic relationship and presence of specific motifs and folds (Figure 1A; [1,25]). Classes A, B and G are single protein units which harbor sites for cofactor binding. In comparison, classes C–F and H exploit external reductases as carriers of reduced flavins. Based on the current status of available sequencing data from plants, it is apparent that plant species only possess class B FMOs.

Class B flavin-dependent monooxygenases are found in all kingdoms of life and typically catalyze hydroxylation of heteroatoms such as nitrogen, sulfur, phosphorus, selenium and iodine [11]. A common feature among Class B FMOs is the strict dependence on a nicotinamide coenzyme (either NADH or NADPH) for the reduction of the flavin cofactor (specifically flavin adenine dinucleotide; FAD). Already prior to substrate binding, the FMO enzyme forms a complex with NADPH, leading to a reduction of the enzyme-bound FAD molecule. The reduced flavin–enzyme complex binds molecular oxygen, resulting in an unstable C4a-(hydro)peroxy flavin intermediate. This state is highly reactive; a so-called "loaded gun". When the enzyme meets a heteroatom-containing substrate (such as N or S) of which it can sterically interact, oxygen is transferred to the substrate. $NADP^+$ and water are then released, and the binding of NADPH starts a new cycle [26]. FMOs are therefore not dependent on an external oxidoreductase protein partner to be catalytically active. Class B FMOs are between 50–60 kDa and harbor FAD and NADPH binding domains [27]. The domain architecture consists of two Rossmann folds 'GxGxxG' (CATH domain 3.50.50.60); one of which binds FAD and the second Rossmann fold binds NADPH (Figure 1A) [27,28].

Class B FMOs have been further classified into three sub-classes [29]: (1) N-hydroxylating monooxygenases (NHMOs), (2) Baeyer-Villiger monooxygenases (BVMOs), and (3) sub-class B flavoproteins (BFMOs, and often referred to flavin-containing monooxygenases) [30]. The enzymatic and phylogenetic diversity between these three sub-classes relates to the position of 49 amino acid residues specifically predicted to interact in cofactor and substrate binding [25]. As this sub-classification is defined by the phylogenetic relationship rather than function, the naming of particular FMOs can create significant confusion, as enzyme function can extend across the different sub-classes and their function-related sub-class name. For example, a single amino acid substitution of a BVMO from *Acinetobacter radioresistens* S13 changed its function to perform N- and S- oxidation reactions instead [31].

In general, plant Class B FMOs have three conserved motifs (Figure 1A): The FAD-binding motif (G*X*G*XX*G), the NADPH-binding motif (G*X*G*XX*G) and the FMO-identifying sequence motif (FXGXXXHXXXY/F) [29]. This FMO-identifying sequence motif is not directly connected to the active site, but rather it acts as a linker section between the FAD and NADPH binding pockets, thus ensuring correct domain rotations and conformational changes [32]. These motifs are present in plant BVMOs and BFMOs known to date, although the motifs can be localized at different positions (Figure 1A). A fourth conserved, but less-defined (F/LA)TGY motif is also present at the C-terminus of N-hydroxylating BFMO across kingdoms [33]. To date, BFMOs have been the predominant FMO sub-class identified within the plant kingdom. Notably, a single BVMO from moss (*Physcomitrella patens*) has been reported [34]. The function of these plant BVMOs and BFMOs are summarized in detail below. As NHMOs are closely related phylogenetically and perform similar enzyme reactions, this subclass is also summarized briefly.

### 3.1. N-hydroxylating Monooxygenases

N-hydroxylating monooxygenases (NHMOs) catalyze the hydroxylation of an amino group with a relatively narrow range of accepted amino substrates [35]. To date, NHMOs have only been identified in bacteria and fungi [36]. The most recognized NHMOs hydroxylate the amino groups of ornithine, lysine or putrescine to form a corresponding hydroxamate. These hydroxamates are precursors in the biosynthesis of siderophores; low molecular-weight iron chelators, which are secreted to then be imported in their iron-loaded form [37]. Iron is involved in several pivotal cellular processes, such as

oxygen transport, respiration, nitrogen fixation, photosynthesis, and synthesis of DNA, fatty acids, and amino acids [38,39].

NHMOs are of emerging interest due to their application in medicine, reprocessing of nuclear fuel, remediation of metal-contaminated sites and the treatment of industrial waste [38,40]. For example, a NHMO was recently characterized to N-hydroxylate aminopropylphosphonate, an intermediate step towards the production of phosphonic acid FR-900098 with potent anti-malaria activity [41].

### 3.2. Baeyer-Villiger Monooxygenases

Baeyer-Villiger monooxygenases (BVMOs) are present in the three domains of life: Archaea, Bacteria and Eukarya, and are considered to be the "ancestors" of all Class B flavoproteins [42]. Large sequence variability of BVMOs can exist, even within species. For example, the red algae genome of a *Rhodococcus* sp. was found to contain 22 different BVMOs [43]. To date, only one BVMO has been identified in a plant (*P. patens*; [34]). BVMOs catalyze the insertion of an oxygen atom between a C–C bond in carbonyl compounds (Baeyer-Villiger oxidation [44,45]). Furthermore, BVMOs can also oxidize C=C bonds to form epoxides and heteroatom-containing molecules [12].

BVMOs have received a high level of attention due to being attractive candidates for "green synthesis" on account of their relatively relaxed substrate profile, high chemoselectivity and preference for mild catalytic conditions. These characteristics facilitate sustainable synthesis platforms, and as a result, there has been an increasing number of reports regarding the engineering of BVMOs [46,47].

### BVMO in Moss (*Physcomitrella Patens*)

A single putative BVMO enzyme from the moss species *P. patens* was identified in 2013, encoded by a continuous open reading frame of the nuclear genome [34]. This was achieved by blasting the FMO identifying sequence "FxGxxxHxxxWP" characteristic to BVMOs against the *P. patens* genome [34,42]. The fingerprint motif for the *Pp*BVMO were not strictly conserved, differing in one amino acid: FxGxxxYxxxWP instead of FxGxxxHxxxWP. To establish the substrate scope of *Pp*BVMO, 46 potential substrates, including aromatic ketones, amines and sulphides, were tested in vitro. The enzyme showed broad catalytic activity against a range of metabolites, including linear and cyclic aliphatic ketones, as well as aryl and aromatic ketones. As an example, the *Pp*BVMO was shown to catalyze the conversion of phenylacetone to benzyl acetate (Figure 2A). To investigate the novel FxGxxxYxxxWP motif, mutants were generated with a Y → H residue substitution. This resulted in an altered substrate profile, with a wider range of substrates being accepted by the mutant enzyme. Furthermore, the mutant enzyme possessed a significantly higher catalytic efficiency [34]. The histidine is considered crucial for catalysis and FAD-binding in BVMOs and other monooxygenase enzymes (e.g., phenylacetone monooxygenase (PAMO) [48]). The physiological role of *Pp*BVMO is unknown; however, it was speculated to be involved in specialized metabolism or modifications of photosynthetic pigments. Interestingly, the BVMO protein of *P. patens* forms a cluster with two other enzymes of bacterial origin: A PAMO and a steroid monooxygenase from *Rhodococcus rhodochrous* [49]; however, the biological and physiological context of this cluster remains unknown. To date, this BVMO is the sole reported BVMO in plants. It is quite possible that more plant BVMOs exist, and will eventually be uncovered as sequencing programs continue to capture a wider array of plant species.

**Figure 2.** Representative plant FMO reactions and their physiological roles. (**A**) The *Pp*BVMO from *Physcomytrella patens* can perform a Baeyer-Villiger C–C cleavage and insertion of oxygen on phenylacetone to form benzylacetate in vitro (amongst other reactions). (**B**–**D**) Characterized FMO reactions from *Arabidopsis thaliana* include oxidative decarboxylation, S-hydroxylation, and N-hydroxylation. (**E**) The characterized *As*FMO1 from garlic (*Allium sativum*) performs an S-oxygenation. Insert: Arabidopsis Yuc6 from also possesses thiol-reductase (TR) activity that is independent of auxin biosynthesis. See text for references.

## 3.3. Class B Flavoproteins (BFMOs)

Class B flavoproteins (BFMOs) generally perform an oxidative reaction resulting in the hydroxylation of heteroatoms such as sulfur or nitrogen [21,25,50]. In mammals and insects, the role of BFMOs has largely been attributed to xenobiotic metabolism, as FMO-mediated hydroxylation renders molecules more polar and facilitates their excretion, detoxification, and transport. Human BFMOs show broad substrate specificity against a number of different molecules, and, most recently, it has

been shown that the BFMOs are in fact metabolizing the molecules that were previously assumed to be metabolized by CYPs [8–10,51].

Based on physiological function and phylogenetic grouping within plant class B FMOs, three distinct clades exist (Figures 1 and 3): The YUCCAs, the S-oxidizing FMOs, and a third clade containing N-oxidizing FMOs.

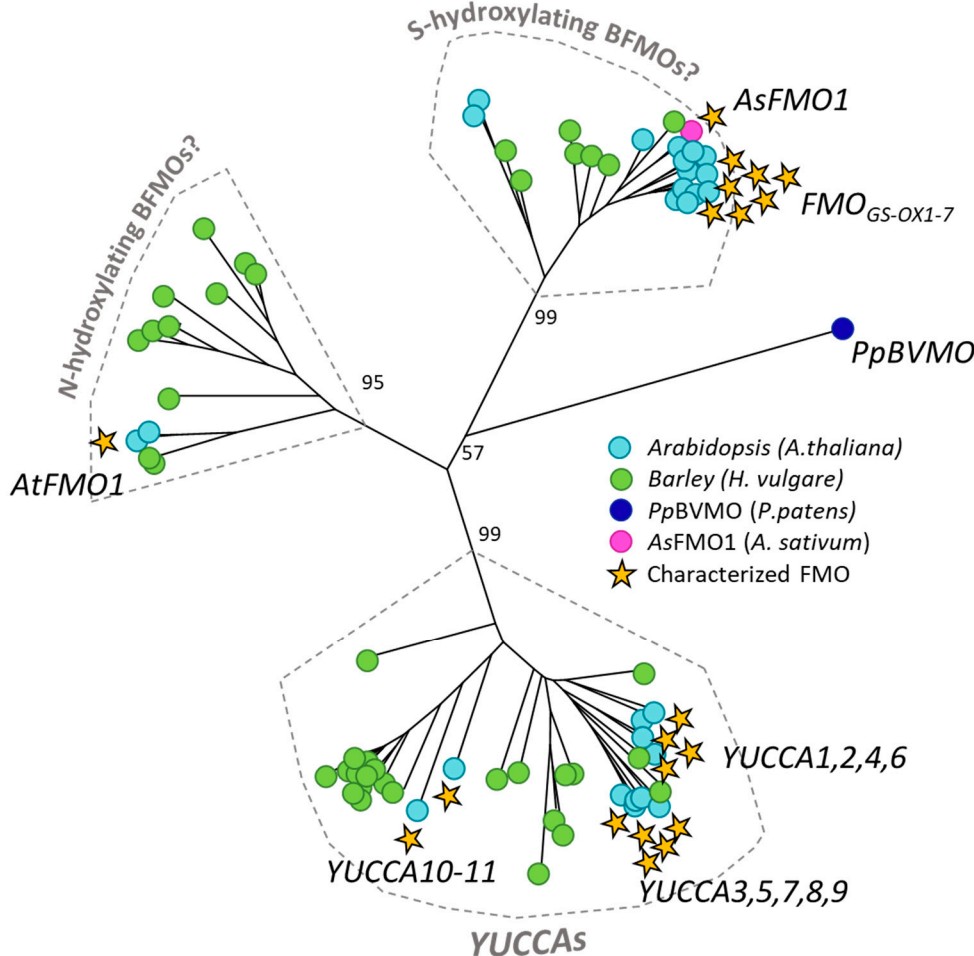

**Figure 3.** Phylogenetic tree of plant FMOs containing all predicted full-length BFMOs from Arabidopsis (*A. thaliana*; cyan; total 29) and barley (*H. vulgare*; green; total 41, predicted from IPK Barley blast server). The tree also includes two additional characterized plant flavoproteins; the functionally characterized S-oxygenating FMO, *As*FMO1, from garlic (*A. sativum*; pink) and the *Pp*BVMO from moss (*P. patens*; dark blue). All characterized proteins are marked with a star. Sequence analyses were conducted using MEGA 7.0. All amino acid sequences were manually inspected before being aligned using ClustalW. The phylogenetic relationship was inferred using the maximum likelihood method based on the JTT matrix-based model and n = 100 replicates for bootstrapping. The analysis involved a total of 74 amino acid sequences. The sequence IDs and resources employed in the phylogenetic analysis, and databases used to extract the sequence data, can be found in Supplementary Materials DataS2.

### 3.3.1. YUCCAs

In 2001, *YUCCA*, an Arabidopsis gene encoding a BFMO was the first FMO to be functionally characterized from plants, and shown to be involved in the production of the hormone auxin (IAA; [52]). Auxin is an essential hormone for plant growth, regulating embryogenesis, as well as leaf, root, flower, vascular and fruit development [53,54]. Furthermore, auxin production has also been shown to be also involved in the response of plants to different stress conditions, including drought

tolerance [55] and shade tolerance [56]. In general, auxin can be biosynthesized from the amino acid tryptophan via four distinct routes involving different metabolic intermediates: tryptamine (TAM), indole-3-pyruvic acid (IPA), indole-3-acetaldoxime and indole-3-acetamide [57]. The initial in vitro characterization of YUCCA demonstrated that it catalyzed the N-hydroxylation of TAM to form N-hydroxy-TAM, an intermediate in the production of IAA. Subsequent in vivo studies later revealed that YUCCA also catalyzes the oxidative decarboxylation of IPA to form IAA ([58–60]; Figure 2B). This *YUCCA* (named *YUC1*), is one of 11 different *YUCCA* genes present in Arabidopsis. Since the characterization of *YUC1* was first reported, the primary research of plant BFMOs over the past 13 years has focused on this clade. Specifically, all 10 remaining *YUCCA*s (*YUC 2–11*) from Arabidopsis have been characterized [52,55,59–63], as well as other plant species, including the orthologue *FLOOZY* from petunia (*Petunia X hybrida*) [64] and fourteen orthologues from rice (*OsYUCCA 1–14*; *Oryza sativa*) [65,66]. *YUCCA* genes have also been characterized from tomato (*Solanum lycopersicum*) [67], maize (*Zea mays*) [68], potato (*Solanum tuberosum*) [69], strawberry (*Fragaria vesca*) [70] and cucumber (*Cucumis sativus* L.) [71] (see [72] for full list). To date, all characterized YUCCAs have been shown to be involved in auxin biosynthesis, catalyzing the oxidative decarboxylation reaction of IPA to form IAA. Catalytically, YUCCAs are not strictly substrate-specific, as they can also catalyze the oxidative decarboxylation reaction of the α-keto acid phenyl pyruvate to form phenyl acetic acid [73] and can N-hydroxylate TAM to form N-hydroxy-TAM [52], in vitro. At present, a full screen of different substrates has yet to be performed and such studies would be highly valuable to understand the catalytic flexibility and promiscuity of YUCCAs.

The high number of *YUCCA*s present within a single genome has been linked to differential expression patterns between tissue types, or in response to environmental stress [56], longevity [16] and auxin homeostasis [74]. Phylogenetic analyses of Arabidopsis *YUCCA*s show that they separate into two sub-clades (Figure 3; [75]). One sub-clade containing five *YUCCA*s (*YUC 3, 5, 7–9*) has been shown to possess tight gene expression patterns related to root development, with *YUC8* and *9* gene expression specifically linked to jasmonic acid signaling for auxin homeostasis [76]. The second sub-clade (including *YUC1, 2, 4* and *6*) has overlapping gene expression patterns during flower development, and double knockouts are required to observe a significant phenotypic effect on development [53,54]. In general, multiple *YUCCA* knock-out combinations are required in order to observe obvious phenotypes, thus it is the current opinion that *YUCCA* genes may be partially redundant. This opinion, however, has also been formed based on an incomplete recognition of the full capability of YUCCA proteins, and the possibility that they may possess dual functionality.

Notably, YUC6 was identified to possess a second distinct protein functionality, independent of auxin biosynthesis [77]. In particular, YUC6 was demonstrated to possess novel NADPH-dependent thiol-reductase (TR) activity which conferred with holdase chaperone properties, enhanced peroxidase activity and improved reactive oxygen species (ROS) scavenging. This enzyme functionality had a direct physiological effect on Arabidopsis plants, whereby overexpression of YUC6 improved drought tolerance [77] and mediated the delay of leaf senescence [78]. Exploration of conserved motifs between the 11 different *YUCCA* genes in Arabidopsis identified a conserved cysteine (Cys-85; Figure 1A), predicted to form part of a redox-active disulfide bond. Site-directed mutagenesis of this residue disrupted the TR functionality of YUC6; however, the ability to catalyze the conversion of IPA to IAA was still maintained. This striking dual role of YUC6 illuminates many possibilities to investigate additional functionalities of other YUCCA and YUCCA-like proteins [79], but also illustrates that caution needs to be taken when interpreting physiological output in knock-out and over-expressing *YUCCA* lines. Despite the overall progress in YUCCA functionality within Arabidopsis and other model plant species, surprisingly little is known about the biochemical mechanisms of these proteins, as well as broader catalytic properties such as substrate specificity.

The biochemical mechanism of YUC6 was established in 2013; a keynote study that remains to be the most exhaustive biochemical characterization of any plant FMO to date. Here, the authors established FAD (not FMN) as the flavin cofactor and demonstrated that a moderately stable C4a

(hydro)peroxy intermediate was formed following the reduction of FAD by NADPH and reaction with dioxygen (Figure 4). It was imagined that a C4a-peroxy intermediate may have been formed instead, based on the mechanistic reaction of the YUC6 (an oxidative decarboxylation) being more similar to that of a BVMO enzyme. However, the YUC6 FAD intermediate aligned with other BFMOs. In contrast to other BFMOs, where the C4a intermediate can possess a long half-life of up to 30 min [80,81], the half-life of the YUC6 C4a intermediate was moderate; approximately 20 s. In the absence of the YUC6 substrate IPA, the C4a-(hydro)peroxy intermediate was shown to decay into oxidized FAD and hydrogen peroxide, a process known as uncoupling (Figure 4). The aforementioned TR functionality of YUC6 (and possibly other YUCCAs) could thereby provide an ingenious self-harboring mechanism to cope with the generation of these ROS.

**Figure 4.** Proposed catalytic mechanism of the plant FMO enzyme, YUC6. YUC6 utilizes FAD and NADPH as cofactors, whereby FAD is reduced by NADPH, and thereafter reacts with dioxygen (red) to form a moderately stable C4a-(hydro)peroxy FAD intermediate. This intermediate reacts with IPA to form IAA, via an oxidative decarboxylation (drawn in blue). In the absence of the substrate IPA, the C4a-(hydro)peroxy intermediate decays into oxidized FAD and hydrogen peroxide, a process known as uncoupling (depicted by the dotted line). The R group denotes ribitol-adenosine-diphosphate. FAD: flavin adenine dinucleotide phosphate; FADH$_2$: flavin adenine dinucleotide; IAA: Indole acetic acid; IPA: Indole-3-pyruvate; NADPH: nicotinamide adenine dinucleotide phosphate. Figure modified from [73].

### 3.3.2. S-oxidizing FMOs

The first sulfur-oxidizing plant FMO, FMO$_{GS-OX}$, was first identified from Arabidopsis and characterized in 2007 [30]. *FMO$_{GS-OX}$* is one of seven highly similar FMOs (*GS-OX 1–7*) present in Arabidopsis [82], forming a phylogenetically distinct subclade of plant FMOs (Figure 3). Specifically, FMO$_{GS-OX1}$ catalyzes the S-oxygenation of the methionine-derived glucosinolate, methylthioalkyl, to methylsulfinylalkyl (Figure 2C). Glucosinolates are specialized metabolites only present in the Brassicales order. Following tissue disruption, glucosinolates are hydrolyzed by endogenous myrosinases to form different hydrolysis products, including isothiocyanates and nitriles. Glucosinolates and their hydrolysis products have diverse biological activities, including chemical

defense against pathogens and herbivores [83]. Furthermore, the methylsulfinylalkyl glucosinolate hydrolysis products have also been shown to have potent cancer-preventative properties [84].

　　Functional characterization of four additional $FMO_{GS-OX}$ genes from Arabidopsis demonstrated that all members are involved in the S-oxygenation of methylthioalkyl glucosinolates. As with $FMO_{GS-OX1}$, $FMO_{GS-OX2-4}$ also catalyzes the formation of methylthioalkyl to methylsulfinylalkyl, independent of the side chain length (C3–C8) [85]. $FMO_{GS-OX5}$ has a more limited substrate specificity, only catalyzing the S-oxygenation of the long chain 8-methylthiooctyl glucosinolate. While not functionally characterized, overexpression of $FMO_{GS-OX6}$ and $FMO_{GS-OX7}$ in Arabidopsis supports a role in the S- oxygenation of short- and long-chain methyltioalkyl glucosinolates [82]. The diversity of $FMO_{GS-OX}$ members in Arabidopsis is an example of how gene duplications can either keep the original function, and display different expression patterns [82] or acquire new functions (e.g., $FMO_{GS-OX5}$) [85]. Interestingly, two closely related $FMO_{GS-OX}$ orthologs were identified recently in the genome of *Barbara vulgaris* (also Brassicales order), a species that apparently lacks methionine-derived glucosinolates [86]. It is hypothesized that these *B. vulgaris* $FMO_{GS-OX}$ may have functionally diversified to catalyze the S-oxygenation of the phytoalexin and anti-fungal specialized metabolite Nasturlexin C, to form Nasturlexin C oxide [86,87]. Beyond the Brassicales order, a BFMO from the $FMO_{GS-OX}$ clade has been characterized from garlic (*Allium sativum*; [88]). This *As*FMO1 performs a highly stereoselective S-oxygenation of S-allyl-L-cysteine, to yield alliin (Figure 2E). Allin belongs to the specialized metabolite class of S-alk(en)yl cysteine sulphoxides, which are responsible for the distinct flavor compounds within the *Allium* (onion) genus, and also possess beneficial properties for human health [89,90].

　　With access to more sequencing data, is it clearly apparent that FMO orthologs of $FMO_{GS-OX}s$ are present in all land plants (e.g., seven members in barley; Figure 3). While their involvement in the biosynthesis of glucosinolates and S-alk(en)yl cysteine sulphoxides is established, the substrates for the remaining FMOs in this clade across the plant kingdom are unknown. Glucosinolates and S-alk(en)yl cysteine sulphoxides are specialized metabolites restricted to specific plant taxa, and are unlikely substrates in other plant species. It could be envisioned that the other members of this class are also performing S-oxygenation reactions, especially as sulfur-containing metabolites are widespread across the plant kingdom, involved in both general and specialized metabolism [91].

### 3.3.3. N-oxidizing FMOs

　　Fourteen years ago, Arabidopsis *At*FMO1 was identified to play a critical role in pathogen resistance and plant immunity [92]. Specifically, *At*FMO1 was shown to induce systemic acquired resistance (SAR), a process whereby a chemical signal is used to communicate a broad-spectrum systemic immune response beyond the point of infection. In later years, an *AtFMO1* knock-out Arabidopsis line "*fmo1*" was shown to accumulate the nonproteinogenic amino acid, pipecolic acid (Pip, [93]), and the generation of a pipecolic acid-induced SAR, specifically depends on a functional *AtFMO1* gene [94]. Despite these clues, the enzymatic function of *At*FMO1 and its specific role in SAR remained unknown until 2018, when two separate laboratories characterized it within months of each other.

　　Based on the hypothesis that *At*FMO1 catalyzes the conversion of Pip to its N-oxidized derivative, state-of-the-art chromatography-mass spectrometry analysis was performed on wild-type and *fmo1* plants, identifying minute levels of N-hydroxy-pipecolic acid (N-OH-Pip) as a pathogen-induced metabolite. Successive in vitro assays using purified recombinant *At*FMO1 protein, confirmed its N-oxidizing functionality [95]. In parallel, an independent study using untargeted metabolic analysis of Arabidopsis *fmo1* seedlings identified an O-glycosylated form of N-OH-Pip as an *At*FMO1-dependent metabolite [96]. Utilization of the agro-mediated transient tobacco (*Nicotiana benthamiana*) expression system confirmed that *At*FMO1 catalyzes the N-hydroxylation of Pip to form N-OH-Pip (Figure 2D).

　　The SAR response and defense signaling cascades are not unique to Arabidopsis, with the SAR-induction pathway suggested to be conserved across vascular plants, although the mechanisms are not fully analogous [97]. This hypothesis is supported by the recent characterization of *AtFMO1*

orthologues in different plant species, which were also shown to catalyze the N-hydroxylation of Pip to form N-OH-Pip [98]. Within the phylogenic clade containing *AtFMO1*, a second but uncharacterized orthologous Arabidopsis is also present, but its function remains unknown. Notably, barley possesses 11 *FMO1*-like members in the same clade (Figure 3), a general pattern for monocots [95]. Functional characterization of these FMOs will reveal if all members in this clade are also N-oxidizing and will provide great insight into if and how common an N-OH-Pip SAR immune response is present across the plant kingdom.

## 4. The Road Ahead for Plant FMO Research

Despite the advances in plant FMO research in recent years, many questions concerning mechanism, structural function, selectivity, physiological role and evolution still remain unanswered. To date, the focus of plant FMO research has primarily centered on model species such as Arabidopsis or rice. Below, we discuss strategies to move ahead both within model and non-model plants, and highlight some challenges.

### 4.1. Identification of Substrates

It was wisely concluded by Schlaich in 2007 that "*without any known substrates for plant FMOs, the field would stall*" [21]. Indeed, the major advancements in plant FMO function since 2007 largely lie in the characterization of FMOs where the substrate could be hypothesized; for example, the characterization of YUCCA orthologues in a number of different plant species [64,65,67–71].

The challenges related to substrate identification are exemplified by the long and combined efforts required to resolve the function of *At*FMO1 [99]. Here, the identification of minute N-OH-pip levels was key towards *At*FMO1 characterization [95,96]. Accordingly, the task of identifying substrates is still a major limitation regarding the advancement of plant FMO research today.

It is envisioned that the major advances in analytical chemistry, specifically the increased capabilities in sensitivity and accuracy, are now at a level which will facilitate the identification of further FMO substrates; even if present in low amounts. The increasing number of large-scale untargeted metabolomic analyses and synergy to other "omic" investigations and a systems biology approach will provide valuable data resources for the mining of potential substrates. Furthermore, advances in technologies such as MALDI-MS imaging enable the ability to visualize the tissue, cellular and subcellular localization of different metabolites [100–102]. If this technique is then also coupled to single-cell RNA sequencing, enzyme activity and function could be pinpointed to a particular time and space, therefore greatly enhance our understanding of plant FMOs.

From an evolutionary perspective, the advances and accessibility in next-generation sequencing technologies will aid the fundamental understanding into the conservation and diversification of FMO function over evolutionary time. Here, large sequencing initiatives such as the 1000 plant transcriptomic project [103], which targeted different non-model and evolutionary diverse plant species, have provided a large and invaluable genetic toolbox. Highly conserved clades of FMOs, conserved from primitive green algae through to higher flowering plants, likely signifies a fundamental role in plant metabolism, thus leading to qualified hypotheses on substrate profile. Similarly, diversified FMOs, such as the comparatively more diversified *AtFMO1* and $FMO_{GS-OX}$ orthologues present in barley (Figure 3), could suggest substrates specific within barley and related monocotyledonous species.

### 4.2. A Need for Structural Insights of Plant FMOs

Crystallization studies provide invaluable insight into the mechanism, function and evolution of enzymes. To date, no plant FMO has been crystalized, and relatively few FMO structures are known, especially within the BFMO class. In 2006 and 2008, the first crystal structures of BFMOs were released, specifically the BFMO flavoprotein from *Schizosaccharomyces pombe* [104] and the *m*FMO from the methylotrophic bacterium *Methylocella silvestris* [105]. It was then not until 2018, that a BFMO from a multicellular organism was crystalized: The ZNO pyrrolizidine alkaloids monooxygenase from the

grasshopper species *Zonocerus variegatus* [106]. Most recently, ancestral reconstructed human FMO1, 3 and 5 were crystalized [107]. Here, the ancestral sequences are 90 percent identical to extant versions and maintain their functions completely.

This establishment of three-dimensional structures of BFMOs guides the intimate understanding of substrate preference and can ultimately aid in the prediction of potential substrates, or the identification of key residues for enzyme function modification. Regarding the reconstructed human FMOs, enzyme structure revealed an apparent substrate tunnel and enables the docking of candidate molecules based on their size and hydrophobicity [107]. Similarly, crystallization of a BVMO facilitated insights in cofactor binding and orientation. A single arginine residue was identified as key in the special reaction mechanisms this enzyme class [32], laying the foundation for further mutagenesis studies.

The link between tertiary structure and plant FMO binding and function has been hypothesized in a few cases, such as the prediction of substrate specificity of FMO$_{GS-OX5}$ [85]. Overall, however, there are very few references to plant FMO structure, or even homology predictions, such as the homology modeling of YUC2 based on the structure of its ortholog FMO in yeast [108].

Notably, all published structures of Class B flavin-dependent monooxygenase suggest they operate in a homo-dimeric organization [32,104–107]. Albeit this oligomerization is not reported directly for plant FMOs, such organization is very plausible and should be considered when plant FMOs are handled (e.g., enzyme purification and screening). Dimerization is interesting from an evolutionary perspective as residues of interaction would, in theory, be tightly constrained. Accordingly, the oligomeric state of plant FMOs should also be taken into consideration in mutagenesis studies.

### 4.3. Optimizing Strategies for Plant FMO Biochemical Studies

Biochemical and metabolic studies on FMOs rely on the ability to purify and/or recombinantly express and isolate as active proteins. Unfortunately, in this regard, eukaryotic class B flavin-dependent monooxygenases are notoriously challenging to handle, with reports on isolated proteins for in vitro experiments relatively scarce in the literature. Successful purification and recombinant expression of plant-specific FMOs are similarly scarce, or with insufficient quantities of recombinant protein achieved for detailed biochemical analysis.

The first plant FMO successfully purified and biochemically characterized was YUC6 from Arabidopsis [73]. The successful expression and purification of this His-tagged protein for biochemical analysis was aided by growing *E. coli* cells at a low temperature and including a high concentration of glycerol in the purification buffers [73]. These technicalities also enabled the co-purification of YUC6 and its FAD cofactor, a critical step towards acquiring an active protein [96], as well as facilitating the tracking of intermediates during the biochemical reactions.

Successful biochemical studies have since been achieved for the *Pp*BVMO [34] and *At*FMO1 [95]. However, in contrast to YUC6, neither a low temperature expression, cold-adapted chaperones, nor fusion protein resulted in effective quantities of active *Pp*BVMO. Active *Pp*BVMO protein could only be purified following addition of excess riboflavin (the FAD-precursor) to the culture medium [34]. Active soluble *At*FMO1 was achieved by using a shorter incubation time (down to 5 h at 28 °C) after induction with IPTG [95], although *At*FMO1 activity was still reported to be low. Only by successful purification of active plant FMOs will enzyme properties such as substrate specificity be established. The limited in vitro analysis conducted on plant BFMOs to date (e.g., GS-OXs and some YUCCAs, see above), in combination with in planta observations, would suggest that plant FMOs are more substrate-specific compared to the very broad substrate profile of detoxifying human BFMOs. Comprehensive screening of different plant FMOs is needed to obtain a greater catalytic understanding of these enzymes.

With the apparent challenges and limitations regarding expressing sufficient active protein for biochemical and metabolic studies, it is important for the FMO community to continue to be open about methodology, especially by sharing techniques across the bacterial, fungal, mammal and plant fields.

### 4.4. What's in a Name? FMO Nomenclature

The field of FMO research is becoming a very dynamic and fast-moving field, and forward navigation within this enzyme class could be greatly strengthened by a cross-kingdom and ordered naming system that would encompass and cleanly organize all FMOs. Until now, FMO nomenclature and the naming of new enzymes is somewhat improvised, and, as a result, the ability to recognize past work in the literature is hampered due to difficulties in distinguishing particular FMO classes, and their phylogenetic and/or possible functional relationships to each other. For example, the Group B sub-classes NHMO, BVMO and BFMO are misleading, as the division in fact relies on phylogeny, and not function, as the sub-class naming indicates. Consequently, optimal synergy and crosstalk relating to research strategies and methodologies (e.g., the limitations of biochemical studies; see Section 4.3) within and between FMO subgroups is at risk, especially as more FMOs are identified and characterized.

The abundant number of FMOs currently named by function, although historically interesting, also creates confusion. For example, the naming of $FMO_{GS-OX}$ (here "GS-OX" corresponds to "glucosinolate-oxidizing") is no longer intuitive, as additional members of the clade are involved in the production of metabolites other than glucosinolates. Similarly, the names "*As*FMO1" (As = *Allium sativum*) and "*At*FMO1" (At = *Arabidopsis thaliana*) may generate misunderstandings, as these two FMOs are not homologous to each other phylogenetically or functionally, nor hold any close resemblance to the similarly named human "FMO1".

The ongoing sequencing of genomes, as well as large-scale transcriptome initiatives across the tree of life, provide substantial material and information by which the FMO community could estimate the total FMO number and establish a structured and searchable nomenclature system to complement the existing names. It could be imagined that an A-level classification system (e.g., class, group, family, and subfamily) may be a viable naming solution, especially as it has proven to be highly useful for the categorization of highly abundant cytochrome P450 genes [109].

## 5. The Unexploited Potential of Plant FMOs for Agriculture and Biotechnology

Significantly, the plant FMOs characterized to date perform novel oxygenation reactions that are crucial steps in hormone metabolism, abiotic and biotic stress resistance, signaling and chemical defense (Figure 2A–E). Furthermore, the notable case of YUC6 also demonstrates that plant FMOs may possess additional and physiologically beneficial functionality, which is independent of their FMO-specific catalytic ability [77]. Together, these roles demonstrate the fundamental importance of FMOs within plant metabolism, and presents many opportunities for future agricultural applications, especially within crop improvement. The oxygenation reactions performed by plant FMOs contribute to an expanding toolbox of monooxygenase enzymes that possess valuable potential for the development of new biocatalysts and the generation of novel bioactive metabolites.

From an agricultural perspective, a number of Arabidopsis FMOs have been specifically expressed in other crop plants with an aim to improve stress performance and to identify analogous functionality. The field has predominantly focused on the exploitation of YUCCAs, based on their physiological role in hormone production, an improved abiotic stress response [110] and the overall need for crop resilience under future adverse climatic conditions [111]. Indeed, when *AtYUC6* was stably transformed into important crop plants such as potato (*Solanum tuberosum*), sweet potato (*Ipomoea batatas*), soybean (*Glycine max*), and poplar (*Populus alba* × *P. glandulosa*), improved abiotic resistance was observed in all cases [112–115]. More recently, the characterization of *At*FMO1 and its critical involvement in SAR and plant immunity has opened up exciting opportunities to identify or engineer analogous defense response signaling in other crop species. The molecular elucidation of *At*FMO1 and identification of N-OH-Pip has recently enabled the examination of an analogous system across multiple plant species [98]. Here it was shown that N-OH-Pip is synthesized by a diverse array of plant species, and that the production of N-OH-Pip also triggers a SAR response. This lays the important ground work to engineer biosynthesis of N-OH-Pip to increase the immunity response of crops to pathogen attack, and improve overall agricultural productivity.

Beyond developing plants for crop improvement, plant FMOs and their pathway partners can also be engineered in non-native hosts for the production of valuable bioactive metabolic products. Namely a synthetic biology approach can be utilized to successfully express multiple genes in concert, either in "plant factories" [116] or cell lines (e.g., *Escherichia coli* or *Saccharomyces cerevisiae*). A key example of this approach is exemplified by the engineering of a 13-gene pathway (including $FMO_{GS-OX}$) in tobacco (*N. benthamiana*), giving rise to glucoraphanin [117]. Glucoraphanin is the major glucosinolate from broccoli (*Brassica oleracea*) and possess high-value chemopreventive properties, as it is converted to sulforaphane upon consumption [84,118]. Studies with animal and cell models have provided evidence that sulforaphane has cancer-preventive properties, which include antioxidant functions, anti-inflammatory properties, apoptosis-inducing properties, and the induction of cell-cycle arrest [119]. The engineering of this complex pathway provides great potential as a means to generate a stable, rich source of glucoraphanin for use in the food and medicinal industry for the benefit of human health. Similar engineering could also be applied from other valuable metabolites, like the S-alk(en)ylcysteine sulphosides from *Allium* species [88,90]. It is conceivable that plant FMOs may be involved in the production of a diverse array of bioactive metabolites that may possess value for society (e.g., pharmaceutical or nutritional value). In addition, it is also possible that with further understanding of biochemical mechanistic and functionality, the rich diversity of plant FMOs could be manipulated, exploited and designed to specifically catalyze desired oxygenation reactions.

From a biocatalytic perspective, FMOs, and monooxygenases in general, constitute an important class of enzymes with huge potential for industrial development [120,121]. The catalytic insertion of a single oxygen atom from molecular oxygen ($O_2$) into an organic substrate—often with high chemo-, regio- and enantio-selectivity—alters its physical and chemical properties, such as polarity, solubility, reactivity, and susceptibility, for further enzymatic modifications [1]. In comparison to other monooxygenases, such as cytochromes P450s, FMOs offer several catalytic advantages, such as their soluble nature, their so-called "loaded gun" reaction mechanism and their autonomy from external oxidoreductase protein partners in order to be catalytically active. While FMOs are relatively underused in industry at present, several bacterial FMOs are exploited for commercial pursuits, for example, in the pharmaceutical and dye industries [20,122]. The relatively limited use of FMOs as biocatalysts today is partly due to the restricted availability of enzymes, both in terms of their expression and isolation [1], although some successful strategies have be employed to address these limitations [107].

A noteworthy constraint for the wide application of FMOs in industry is the expensive consumption of NADPH or $NADP^+$. This challenge is general for all $NADPH/NADP^+$-consuming monooxygenases, thus a rising number of innovative enzymatic, chemical, electrochemical and phytochemical regeneration approaches are continually being developed to ensure the regeneration of cofactors [123,124]. FMOs have been coupled to enzymatic partners in $NADPH/NADP^+$ recycling systems since 1991, when a BVMO was used in combination with an alcohol dehydrogenase [125]. Here, an oxidizing $NADP^+$-dependent alcohol dehydrogenase produced NADPH, for the NADPH-consuming BVMO reaction. Furthermore, it was established that the selectivity of this BVMO is not altered by these regeneration conditions. In the opposite way, FMOs also possess significant potential to be developed as NADPH oxidases, regenerating the cofactor $NADP^+$ for use by other catalysts. At present, very few $NADP^+$ regeneration catalysts are available [124,126]. The biocatalytic potential of FMOs in this regard is due to their ability to uncouple from $NADP^+$ in the absence of a substrate, following the reduction of FAD and reaction with molecular oxygen (Figure 4). A key example of this application is exemplified by the thermostable BVMO (specifically a PAMO) from *Thermobifida fusca*. Mutation of a single amino acid (C65D) in this BVMO resulted in a more efficient uncoupling rate (i.e., exhibiting strong NADPH oxidase activity), whilst maintaining its high thermostability, narrow substrate specificity and solvent tolerance [127]. It is envisioned that plant FMOs could also constitute candidates for $NADP^+$ regeneration enzymes, especially as they as possess a relatively faster decay and uncoupling of the C4a-hydroperoxy-intermediate (and

presumably NADP$^+$ production as reported for YUC6 [73]), and narrow substrate profile compared to human BFMOs.

Beyond the challenges currently being overcome for the development of FMOs in general, there remain limitations relating to plant FMOs specifically. This primarily relates to our limited understanding of biochemical and structural mechanists; for example, our limited knowledge on substrate and catalytic specificities (see Section 4). These limitations, however, offer many opportunities to expand our understanding on plant FMO functionality at the protein and organism level. Overall, from both a biological and industrial point of view, plant FMOs possess many advantageous properties. At present, they are a substantially underutilized class of enzymes, but hold a rich potential which can be exploited for the discovery of new activities and novel bioactive metabolites. Accordingly, it is envisioned that in the future, plant FMOs can be optimized and utilized for a wide range of commercial applications within the pharmaceutical, biotechnology and agri-business sectors.

## 6. Conclusions

Plant FMOs are a diverse class of enzymes that catalyze crucial steps within hormone metabolism, pathogen resistance, signaling and chemical defense. Despite these important roles, the identification of new reactions and further understanding of FMO functionality over the past decade have remained relatively limited and are restricted to a small number of plant species. Technological advances, especially within the field of analytical chemistry and the ability to identify small molecules with high sensitivity and accuracy, will undoubtedly facilitate the characterization of new plant FMO roles, as recently demonstrated for *At*FMO1 [95,96]. The evolutionary diversity of plant FMOs provides a remarkable untapped resource of "green" biocatalyst. Based on their known essential roles in plant metabolism, it is foreseeable that plant FMOs could provide unique breeding targets for agricultural pursuits. Furthermore, the unique capabilities of plant FMOs could be exploited or modified to generate valuable biocatalysts that perform unique reactions, relevant for the biotechnological and pharmaceutical industries.

**Supplementary Materials:** The following are available online at http://www.mdpi.com/2073-4344/10/3/329/s1, The data used to build Figure 1B were extracted from resources listed in DataS1. Sequences used to build tree (Figure 3) are available in DataS2.

**Funding:** This research was supported by a Novo Nordisk Emerging Investigator Grant (Grant No. 0054890), a VILLUM Foundation Young Investigator Grant (Project No. 13167), and a Sapere Aude Postdoctoral Fellowship (Grant No. 6111-00379B) awarded to EHJN.

**Acknowledgments:** We wish to thank webdwarf.dk for kindly providing the plant images featured in Figure 2.

**Conflicts of Interest:** The authors declare no conflict of interest.

## Abbreviations

| | |
|---|---|
| BFMO | Class B Flavin-containing monooxygenase |
| BVMO | Baeyer-Villiger monooxygenase |
| CYP | Cytochrome P450 monooxygenase |
| FAD | Flavin Adenine Dinucleotide |
| IAA | Indole acetic acidIPA: Indole-3-pyruvate IPTG: Isopropyl β-D-1-thiogalactopyranoside |
| NADH/NADPH | Nicotinamide coenzyme |
| N-OH-Pip | N-hydroxy pipecolic acid |
| NHMO | N-hydroxylating monooxygenase |
| PAMO | Phenylacetone monooxygenase |
| Pip | Pipecolic acid |
| ROS | Reactive oxygen species |
| SAR | Systemic acquired resistance |
| TAM | Tryptamine |
| TR | thiol-reductase |

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
