# Peer review of "The “Green” FMOs: Diversity, Functionality and Application of Plant Flavoproteins"

_catalysts, doi:10.3390/catal10030329_

Round 1

Reviewer 1 Report

This work compiles the last advances about plant FMOs in last decade since a previous review: https://doi.org/10.1016/j.tplants.2007.08.009. The provided information is valuable for the field and the summary of the main hits is good. The references are adequate and proportionate to the information. However, the style and sometimes also the grammar are not good enough for direct publication. It seems that the authors did not read the last version of the manuscript before submission, since many full stops, commas, tabs and spaces are missing, and they must be reviewed and corrected. Moreover, the tittle does not represent the information of the manuscript. From the tittle, one can expect that 40-50% of the manuscript will explain the biocatalytic potential of flavoproteins in plants. However, this information is reduced at section 5 in less than one page. I suggest either changing the tittle or giving a better balance between the function and the biocatalytic potential.

Minor concerns:

The citations seem to be in 2 different styles along the manuscript. Some of the citations are numbers, but others are author name and year. Please, write all the citations as reference numbers following the publication requirements of the journal. Check the spelling of all the paragraph as well as full stops, commas, tabs at the beginning of the paragraphs, lower/upper cases and spaces along the manuscript. The word “cofactor” is spelled also as “co-factor”. Although both are correct, choose one word and replace it along the manuscript to avoid misunderstandings. Wrong spelling of “Pipecolic acid” along the manuscript. Check it. Data S1 has not been provided with the main manuscript for reviewing. It is mentioned in line 241 and 489. Line 28: Wrong spelling. Please, change “region-“ by “regio-”. Line 30: The sentence: “Accordingly, FMOs are important oxygenase enzymes […]” is very redundant. I suggest either to change the word “oxygenase” by “oxidoreductase” or just delete it. Figure 1(B): The names of the species must be written in italic and lower case, e.g. change “ Thaliana” by “A. thaliana”. Line 72: Check the grammar. Remove “are”. Line 85: Check the grammar. The sentence must be replaced by “This incorporation of oxygen alters the metabolite physical and chemical properties such as polarity […]”. Line 486: Wrong spelling of the words “Pipecolic acid”. Please, change it. Line 131: Remove “as”. Lines 146-147: The sentence “members of Class B BFMO […]” does not make sense when BFMO has already been described as class B of FMO. Please, remove the redundancy. Line 155: I guess the authors means “lysine”. Please, correct it. Line 157: Remove “later”. Line 157: Remove “in”. Lines 169-170: For a better understanding, change the sentence to: “To date, only one BVMO has been identified in a plant ( patens)”. Line 179: The name of the organism must be in italic. Line 182: “BVMO FMO identifying sequence […]” is very redundant. Line 188: Wrong grammar. Pease change to “benzyl acetate is”. Lines 194-195: The name of the organism must be in italic. Line 208: I guess the authors mean “Figures”. Please, correct it. Line 245: Correct the grammar mistake. The correct sentence is: “One sub-clade containing five YUCCAs (YUCCA 3, 5, 7-9) has been shown […]”. Line 261: The sentence refers to Figure 3C, not 3C. Please, correct it. Line 289: Figure 3F does no exist. It must be Figure 3E. Please, correct it. Line 295: The sentence is not clear and very difficult to understand. Please, change it. Line 305: I suggest to change the concept of “nonprotein” by “non-proteinogenic”. Lines 373-376: The sentence is not clear. Check the sentence. Line 425: The correct sentence should be: “the abundant number of FMOs […]” Line 454: The correct sentence should be: “Nowadays, the relatively limited use of FMO as a biocatalyst is […]”. One of the most promising biocatalytic applications of FMOs is their use as a cofactor-recycling system in multienzyme systems. FMOs have been coupled to other enzymes (dehydrogenases) in order to recycle the redox cofactor NAD(P)H/NAD(P)+, removing the need for external addition of expensive cofactors. This wide-range application should be mentioned in section 5.

Reviewer 2 Report

I found the review proposed by Thodberg et al. really interesting. Besides treating an argument of general interest, it gives perspectives to the field, highlighting what is missing, rather than merely making a list of achievements.

In particular they showed the possible applications of Flavoproteins from plants to catalyze oxidative reactions. 

To make such reading appealing to chemists and biochemists, however, it would be good to report the reaction schemes of the possible transformations catalyzed by Plant Flavoproteins, with the efficiency and selectivity obtained until now in different cases.

It is evident the expertise of authors in plant science, but the chemical view necessary "to go green" in catalysis is missing. Catalysts is anyway a journal read mainly by chemists, even if not exclusively. 

I strongly encourage the authors to enlarge the analysis of this review on a more chemical point of view.

Few typo mistakes, like in line 160, in in medicine.

Line 261, may be there is a mistake referring to figure 3D. "Specifically, FMOGS-OX1 catalyzes the S-oxygenation of the 260 methionine-derived glucosinolate methylthioalkyl to methylsulfinylalkyl(Figure 3.D)." Figure 3D is actually showing the reaction of pipecolic acid.

Reviewer 3 Report

In cannot be stated, based on the abstract, what is the reason for reviewing the field. 

The paper mentiones a broad field of data, however fails to review it critically. Also the reference list is quite long, but the revised papers are not up to date. You can only find few papers from the last three years. I am worried about the topic significance. You have declared:

"This review aims to summarize the diversity and role of characterized plant FMOs. Specifically,we will highlight the field’s progress since the detailed review by Schlaich in 2007; a review made when the role of YUCCAs and glucosinolate-related FMOs was just established [21]. With the large
developments in genome and transcriptome sequencing, this review will also provide an evolutionary context of FMO diversification. Lastly, we seek to highlight the opportunities of FMO research for the next 10 years to maximize the potential of FMOs for future green applications"

Unfortunately there are a lot of similarities beetwen Schlaich 2007 and the given review. 

It was stated, that  Class B flavin-dependent monooxygenases have three conserved motifs, then four were described. Please refer. 

The whole 3rd chapter seems to describe the well known facts, please give comments, as you are preparing a review.

Please specify the source of figures 1, 2 and 3. 

The main aims given in the abstacts section are not addressed in the conclusion section, you should explain how were your findings met.

Several typo occured:

title - Biocatalytic

line 11 - kingdoms?

line 160 - in in 

Round 2

Reviewer 3 Report

Thank you for your reply. The given corrections are convincing and improve the papers understanding.